# The impact of working hours on pregnancy intention in childbearing-age women in Korea, the country with the world's lowest fertility rate

Taewook Kim *

Presidential Committee on Aging Society and Population Study, Seoul Government Complex, Seoul, Republic of Korea

* ray0601@snu.ac.kr

**Data Availability Statement:** All KNHANES dataset files are available on the following KNHANES website: http://knhanes.cdc.go.kr

**Funding:** The author received no specific funding for this work.

## Abstract

This study aimed to assess factors affecting pregnancy intention among women of reproductive age in Korea. We analyzed data from the Korean National Health and Nutrition Examination Survey (KNHANES), a population-based survey that included 22,731 women aged 15–49. As age was associated with birth year and was found to be a confounding factor in the analysis of participants' characteristics, we used propensity score matching to assess the characteristics of pregnant women compared with non-pregnant women of the same age and birth year. We also employed the XGBoost machine learning model to identify the most important factors related to pregnancy intentions. Our feature importance analysis showed that weekly working hours were the most significant factor affecting pregnancy intentions. Additionally, we performed cluster analysis and logistic regression models to determine optimal weekly working hours. Cluster analysis identified participants into three distinct groups based on their characteristics, indicating that the group with an average of 34.4±12.9 hours per week had the highest likelihood of becoming pregnant. Logistic regression was used to analyze the odds of pregnancy for every 5-hour increase in weekly working hours. The results of logistic regression indicated that women who worked between 35–45 hours per week had higher odds of pregnancy, with significant odds ratios of 2.009 (95% confidence interval: 1.581–2.547, p < .001) for 40–45 hours per week and 1.450 (95% confidence interval: 1.001–2.040, p < .05) for 35–40 hours per week, compared to women working other hours. In Korea, the standard workweek is typically 40 hours; however, Koreans often work considerably longer hours, with the second-highest number of working hours among OECD countries in 2022. This study suggests that strict monitoring of working hours and expansion of telecommuting for childbearing-age women are important factors in increasing the fertility rate in Korea.

**Competing interests:** The author have declared that no competing interests exist.

## Introduction

The fertility rate in OECD (The Organization for Economic Cooperation and Development) countries has decreased from 2.84 in 1970 to 1.59 children per woman in 2020, with some countries experiencing rates below the required level of 2.1 for maintaining a stable population. Low fertility accelerates the aging of society and causes multiple problems, such as a decrease in labor productivity and a burden on the younger generation through taxes and other obligations [1, 2]. Prior to 1984, Korea's total fertility rate was higher than the OECD average, with a rate of 1.74. However, the rate has continuously declined, and Korea entered a state of "lowest-low fertility" (less than 1.3) with a rate of 1.18 in 2002. In 2022, Korea's total fertility rate was at a record low of 0.78 [3], meaning that 78 babies are expected to be born every 100 women's lifetime. According to the "World Population Prospects 2022" review, Korea's total fertility rate has been ranked 198th of 198 countries for four years and is the only country with a fertility rate below 1.0 in 2022.

In Korea, advanced maternal age refers to 35 years or older, which is associated with increased risk during pregnancy, such as gestational diabetes, preterm births, and congenital defects in newborn babies [4]. The proportion of women who intentionally or unintentionally delay marriage and pregnancy beyond the age of 35 has increased in Korea. The percentage of pregnancies in women of advanced age has increased from 6.2% in 1999 to approximately 35.7% in 2022, and the average age of first childbearing has increased from 27.6 years in 1993 to 31.6 years in 2017 [5–7].

Previous studies have shown that advanced maternal age and low fertility rates are complex and multifactorial phenomena that can be influenced by factors such as low income status, educational aspirations, and career goals [8, 9]. However, studies on the relationship between work-related factors and pregnancy are limited. As the proportion of women in the workforce increases in modern society [10, 11], we focused on the effects of women's working conditions, particularly employment status, occupation type, and weekly working hours on pregnancy. In this study, we aimed to identify the key factors that influence the intention to become pregnant among women of childbearing-age in Korea, and to determine the optimal working hours that increase the likelihood of preparing for pregnancy.

## Materials and methods

### Study population

Our study recruited female participants from the Korea National Health and Nutrition Examination Survey (KNHANES) conducted annually from 2007 to 2020. The KNHANES is a cross-sectional survey that evaluates the health and nutritional status of Koreans since 1998. The Korea Disease Control and Prevention Agency (KDCA) conducts the KNHANES annually and includes a representative sample of non-institutionalized civilians in Korea. Approximately 10,000 individuals are sampled annually for the survey. The KNHANES consists of three surveys: a health interview, health examination, and a nutrition survey. Our study utilized the health interview and health examination surveys to investigate the characteristics of women of reproductive age.

Fig 1 illustrates our participant selection process, which began with 61,540 female participants from the KNHANES 2007–2020. We excluded participants with missing or inaccurate data (N = 3,226) and those who were not within the childbearing age range (N = 35,583). Since the definition of the total fertility rate is based on women of reproductive age (15 to 49 years) [3, 7], our study specifically focused on women within this age range. Eventually, we analyzed

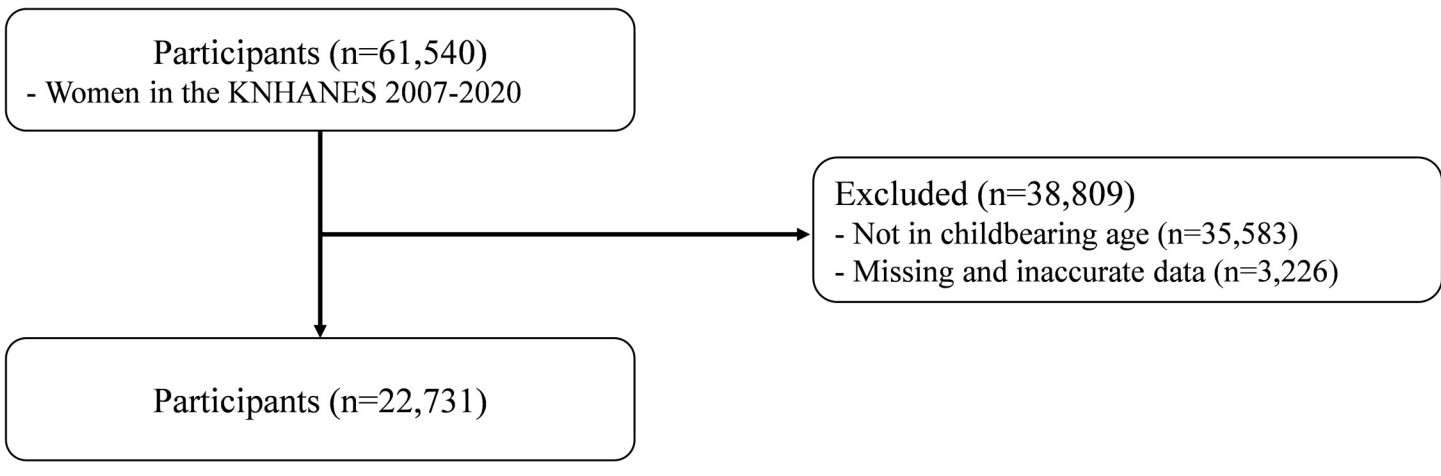

**Fig 1. Flow diagram of study participants in KNHANES.**

data from 22,731 female participants aged 15–49, who answered a set of questions related to their maternal, demographic, and socioeconomic conditions.

## Ethical considerations

This study was based on the KNHANES database, which was approved by the Institutional Review Board (IRB) of the KDCA. We obtained exemption for informed consent from the Public Institutional Review Board Designated by the Ministry of Health and Welfare (IRB No. 2023-0061-001, Approval No. P01-202302-01-017).

## Statistical analysis

The participants' baseline characteristics were categorized into the following age groups: 15–19 years, 20–29 years, 30–39 years, and 40–49 years. Continuous variables, such as employment rate, were presented as mean and standard deviation and compared using the Student's t-test. On the other hand, categorical variables, such as occupation, were presented as percentages and numbers and were compared using the chi-squared test. We set statistical significance at $P < .05$. Additionally, a linear regression analysis was performed to examine the relationship between birth year and age at first birth of participants.

## Propensity score matching for characteristics of pregnant women

The descriptive analysis of baseline characteristics and the linear regression between the age at first childbirth and birth year of women indicated the need to reduce bias for pregnancy tendencies, revealing a significant association among birth year, age, and participant characteristics. For instance, women in the 40–49 age group exhibited higher household income, and women from earlier birth years, indicating the older generation, displayed a tendency to have pregnancies at an earlier age. In addition research has shown that the social environment in which women live can influence their intention to become pregnant. For example, older women may experience pressure from their peers and society to become pregnant as early as possible [12, 13]. Therefore, we used the propensity score (PS) matching method to reduce the confounding bias caused by social conditions. PS is a conditional probability that a person belongs to a specific group, as determined by predictor variables or covariates in the model. The PS matching method pairs participants with similar propensity scores, which helps

balance the distribution of confounding variables between the two groups. We adopted the nearest neighbor method to match the two groups and used a PS caliper to impose a tolerance level on the maximum PS distance to prevent poor matches. We set a high matching ratio between the groups to increase the sample size for analysis. Standardized mean difference (SMD) was used to assess the quality of covariate matching. A SMD less than 0.1 was considered well-paired, indicating a significant reduction in confounding bias.

We examine the major factors associated with advanced maternal age in childbearing women. To investigate the relationship between participants' characteristics and age at pregnancy, we divided them into two groups based on their age at the time of pregnancy: one group included women under the age of 35 years, while the other included those who were of advanced maternal age. We performed PS matching to account for the confounding bias of cultural differences over time. Following matching for birth year, we compared participants' characteristics using the Student's t-test for continuous variables and the chi-squared test for categorical variables. Furthermore, we assessed the major differences between pregnant and non-pregnant women. Participants were classified into two groups based on their pregnancy status, and we adjusted for the confounding bias of cultural differences by performing PS matching for birth year and age. After PS matching, we compared the characteristics of the two groups using Student's t-test for continuous variables and the chi-square test for categorical variables.

## Factors influencing pregnancy intention using machine learning method

We employed the recently developed Extreme Gradient Boosting (XGBoost) algorithm to identify the key factors influencing pregnancy, which has shown to increase the predictive power of models. The XGBoost model for predicting pregnancy intention involved two steps: (1) employing the grid search technique to optimize the hyperparameters of XGBoost and (2) repeating the optimal XGBoost model 100 times to determine the feature importance and model prediction performance.

Initially, we used the grid search technique to optimize the hyperparameters of XGBoost. The grid search involved varying the maximum depth of a tree (2, 4, 6, 8, and 10), learning rate (0.001, 0.0025, 0.005, 0.01, 0.02, 0.04, 0.06, 0.08, 0.1, 0.2, and 0.5), gamma (0, 0.1, 0.5, and 1), subsample ratio of the training instance (0.75 and 1), minimum sum of instance weights (1, 2, and 3), and subsample ratio of columns (0.75 and 1). The root mean square error (RMSE) of each model was compared, and the model with the minimum RMSE was chosen as the optimal model. This resulted in a maximum tree depth of 2, learning rate of 0.1, gamma of 0.5, subsample ratio of 0.75 for training instances, minimum sum of instance weight of 1, and subsample ratio of 1 for columns

Subsequently, we used the optimal hyperparameters to employ XGBoost to predict pregnancy in women of childbearing-age based on their characteristics. To evaluate the model's performance, 70% of the participants were used as the training set, and the remaining 30% were used as the test set. The optimal XGBoost model's performance was assessed by calculating the area under the receiver operating characteristic curve (AUC) based on the Youden Index. An AUC of 0.5 was considered null accuracy, while a higher AUC was considered as better accuracy. We also evaluated other classification evaluation metrics, such as accuracy, precision, recall, and f1 score, to assess the predictive power of the characteristics in the XGBoost pregnancy prediction models.

We estimated the feature importance of each variable by calculating the "gain" of the feature in the pregnancy prediction tree branches. Variables with higher gains were considered more

important in the XGBoost pregnancy prediction model. To obtain reliable estimates, we repeated all analyses 100 times and calculated the 95% confidence interval.

## Cluster analysis and logistic regression to determine optimal working hours

We evaluated the optimal weekly working hours for pregnant women using two methods. First, we conducted a cluster analysis to identify cluster characteristics and examined it with a high proportion of pregnant participants. We computed the dissimilarity values among the participants' characteristics and used the agglomeration method with complete linkage to specify each cluster. Each leaf in the dendrogram corresponded to one participant, and participants closer to each other on the upper side of the dendrogram were considered similar. Based on the dendrogram, we assessed three clusters and compared their characteristics.

To assess the relationship between weekly working hours and the likelihood of pregnancy, we calculated odds ratios for multiple sets of weekly working hours using logistic regression models. We grouped the participants based on their weekly working hours, with each group consisting of a 5-hour range. We then evaluated the performance of logistic regression models for each 5-hour range group, assessing the model with the lowest Akaike Information Criteria (AIC) value and the highest odds ratio for pregnancy (p < .05).

We conducted all analyses using R version 4.2.2 and multiple R libraries, including "xgboost," "car," and related packages on the Windows 10 operating system.

## Results

### Descriptive analysis of participants

We analyzed the characteristics of the participants in our study. Table 1 provides an overview of the age distribution of the participants, with four groups analyzed: ages 15–19, 20–29, 30–39, and 40–49. The data showed significant differences among the groups for several factors, including household income, employment rate, occupation, weekly working hours, educational level, and residence (p < .001). Household income showed a tendency to increase as the age group progressed from 15–19 to 40–49 (p < .001). The participants in the 30–39 age group had the highest education level, however, they also had a higher proportion of unemployment and worked fewer weekly hours compared to other age groups (p < .001).

### Linear regression between age at first birth and birth year of childbearing women

The results of the linear regression model are presented in Fig 2, which suggests that the age when women had their first child increased as the birth year of the participants advanced. These findings imply that the more recent generations tend to delay pregnancy until later ages.

### Characteristics of pregnant and non-pregnant women after controlling confounding variables

To investigate factors associated with delayed pregnancy, we divided participants into two age groups: <35 years and ≥35 years (Table 2). Birth year was matched between the two groups to control for cultural differences over time. Following PS matching, we found that the SMD was -0.008, which was below the cutoff value of 0.1 for well-matched pairs. The results of the PS matching are presented in Table 2. Participants who became pregnant at age ≥35 years worked significantly more hours per week than those who became pregnant at age <35 years (p-

Table 1. Descriptive analysis of childbearing-age women.

| | | Age 15–19 | Age 20–29 | Age 30–39 | Age 40–49 | p-value |
|---|---|---|---|---|---|---|
| | | (N = 2,166) | (N = 4,771) | (N = 7,716) | (N = 8,078) | |
| Age (year) | | 17.0±1.4 | 24.8±2.9 | 34.9±2.8 | 44.5±2.9 | < .001 |
| Birth year | | 1996.9±3.8 | 1988.8±5.1 | 1978.1±4.7 | 1969.1±4.8 | < .001 |
| Household income | | 1,294 (59.7%) | 3,222 (67.5%) | 5,306 (68.8%) | 5,599 (69.3%) | < .001 |
| The top 50% | | | | | | |
| Employment rate | | 371 (17.1%) | 2,809 (58.9%) | 3,887 (50.4%) | 5,054 (62.6%) | < .001 |
| Occupation | White-collar worker | 297 (13.7%) | 834 (17.5%) | 1,327 (17.2%) | 2,804 (34.7%) | < .001 |
| | Blue-collar worker | 73 (3.4%) | 1,963 (41.1%) | 2,549 (33.0%) | 2,243 (27.8%) | |
| | Unemployed | 1,796 (82.9%) | 1,974 (41.4%) | 3,840 (49.8%) | 3,031 (37.5%) | |
| Weekly working hour | | 7.2±14.0 | 28.7±20.6 | 22.0±21.9 | 27.2±22.8 | < .001 |
| Education level | | 3 (0.1%) | 2,651 (55.6%) | 4,733 (61.3%) | 3,260 (40.4%) | < .001 |
| Bachelor's degree or more | | | | | | |
| Residence | | 967 (44.6%) | 2,448 (51.3%) | 3,572 (46.3%) | 3,725 (46.1%) | < .001 |
| Metropolitan city | | | | | | |
| Age at first birth (year) | | 16.5±0.7 | 24.4±2.8 | 27.9±3.6 | 26.9±4.0 | < .001 |
| Current pregnancy | | 0 (0%) | 277 (5.8%) | 656 (8.5%) | 28 (0.3%) | < .001 |

* All variables are mean ± SE (standard error) or frequency %

value < 0.05). However, there were no significant differences in other characteristics including household income, occupation, educational level, and residence.

We divided participants into two groups based on their current pregnancy status to identify the factors associated with pregnancy intention of childbearing-age women. To reduce the confounding bias caused by differences in age and cultural factors, we matched the participants' age and birth year between the two groups. Table 3 presents the results before and after PS matching for the characteristics of women in the current pregnancy groups. We found that the age and birth year were well balanced after PS matching (SMD < 0.1). There was no significant difference in the household income between pregnant and non-pregnant women. However, pregnant women tended to have higher levels of education (p-value <0.001). The proportion of unemployment was higher, while the proportion of white-collar workers was lower, and the average weekly working hours were less compared to non-pregnant women (p-value <0.001).

## Key predictive factors for pregnancy of childbearing-age women using the XGBoost machine learning model

The XGBoost results for predicting pregnancy showed that weekly working hours was the most important factor on pregnancy intention (gain = 0.378±0.002, as shown in Fig 3). The XGBoost model had an AUC of 0.675±0.001, accuracy of 0.516±0.004, precision of 0.166±0.001, recall of 0.783±0.005, and an f1 score of 0.274±0.001.

## Optimal weekly working hours to increase pregnancy rates in women of childbearing-age

Table 4 presents the baseline characteristics of the three clusters identified in this study. Three distinct phenotypes were identified: cluster 1 included pregnant women with a high household income, blue-collar workers, high education level, and low weekly working hours; cluster 2 included white-collar workers and residents outside metropolitan cities; and cluster 3 included

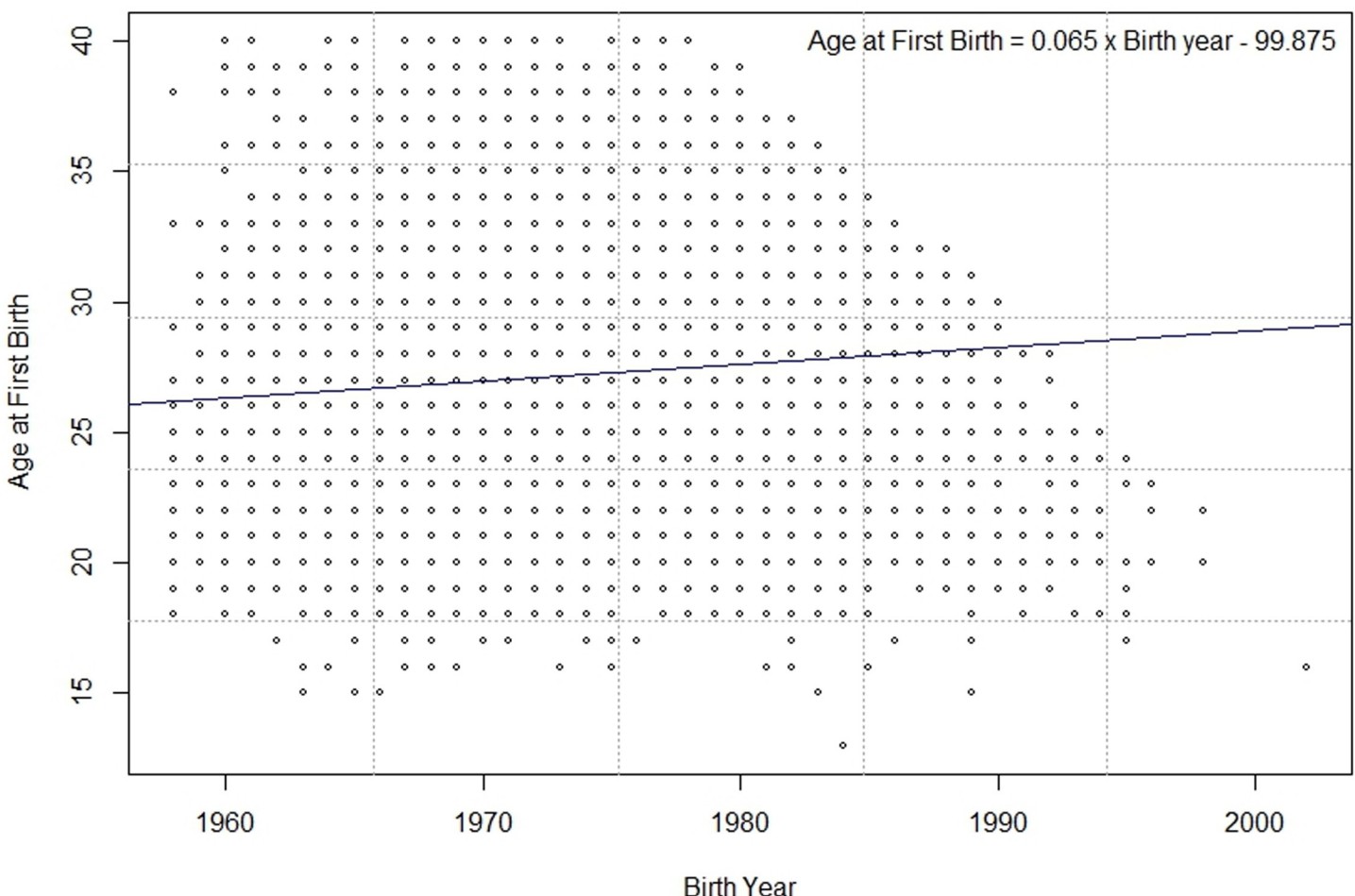

**Fig 2. The relationship between age at first birth and birth year of women with childbirth experience.**

white-collar workers with high weekly working hours. The results showed that cluster 1 was associated with pregnancy, referring to most of the 274 pregnant participants out of 283 pregnant participants. The data also revealed that cluster 1 showed a few weekly working hours (34.4±12.9) compared with cluster 2 (65.0±7.9) and cluster 3 (96.2±8.8), which indicated that weekly working hours was an important factor in determining childbearing women who are willing to become pregnant (p-value <0.001).

A logistic model was used to determine the optimal weekly working hours for predicting pregnancy. The results of this analysis are presented in Table 5, which shows the odds ratios, p-values, and AIC values for each weekly working hours category, sorted by AIC. Our findings revealed that working 40–45 hours per week was associated with an increased probability of pregnancy, with an odds ratio of 2.009 (95% confidence interval 1.581–2.547, p-value<0.001) and a well-fitting model with the lowest AIC. This means that the chance of pregnancy with weekly working hours of 40–45 hours was approximately 2.009 times than the chance of pregnancy with weekly working hours of less than 40 hours or more than 45 hours. Furthermore, the analysis revealed that working 35–40 hours per week was significantly associated with an increased probability of pregnancy (odds ratio 1.450, 95% confidence interval 1.001–2.040, p-value <0.05). Conversely, working less than 35 hours (30–35 hours) and over 60 hours (60–65

**Table 2. PS matching for characteristics of pregnant women between age groups.**

| | | Before PS matching (SMD of birth year = -0.973) | | | After PS matching (SMD of birth year = -0.008) | | |
|---|---|---|---|---|---|---|---|
| | | Pregnant women age<35 (N = 910) | Pregnant women age≥35 (N = 51) | p-value | Pregnant women age<35 (N = 152) | Pregnant women age≥35 (N = 51) | p-value |
| Birth year | | 1980.6±4.7 | 1976.5±4.2 | < .001 | 1976.6±4.1 | 1976.5±4.2 | .901 |
| Age | | 31.5±3.7 | 37.7±2.0 | < .001 | 31.3±4.2 | 37.7±2.0 | < .001 |
| Household income The top 50% | | 604 (66.4%) | 37 (72.5%) | .448 | 109 (71.7%) | 37 (72.5%) | 1.000 |
| Occupation | White-collar worker | 44 (4.8%) | 3(5.9%) | .610 | 6 (3.9%) | 3 (5.9%) | .055 |
| | Blue-collar worker | 218 (24.0%) | 15 (29.4%) | | 23 (15.1%) | 15 (29.4%) | |
| | Unemployed | 648 (71.2%) | 33 (64.7%) | | 123 (80.9%) | 33 (64.7%) | |
| Weekly working hours | | 17.3±21.2 | 17.9±20.7 | .824 | 11.2±18.6 | 17.9±20.7 | < .05 |
| Education level Bachelor's degree or more | | 608 (66.8%) | 34 (66.7%) | 1.000 | 93 (61.2%) | 34 (66.7%) | .594 |
| Residence Metropolitan city | | 420 (46.2%) | 24 (47.1%) | 1.000 | 73 (48.0%) | 24 (47.1%) | 1.000 |

\* All variables are mean ± SE (standard error) or frequency %; SMD showed the balance of covariate distribution between two matched groups, indicating SMD<0.1 was considered a sign of good balance.

hours) was associated with a decreased probability of pregnancy, suggesting that there is a range of optimal working hours for predicting pregnancy and that working too few or too many hours per week may have a negative impact on a woman's chances of becoming pregnant.

**Table 3. PS matching for characteristics of women between current pregnant groups.**

| | | Before PS matching (SMD of age = -0.691) (SMD of birth year = 0.311) | | | After PS matching (SMD of age = 0.058) (SMD of birth year = -0.065) | | |
|---|---|---|---|---|---|---|---|
| | | Pregnant women (N = 961) | Non-pregnant women (N = 21,770) | p-value | Pregnant women (N = 7,676) | Non-pregnant women (N = 961) | p-value |
| Birth year | | 1980.4±4.8 | 1978.9±10.6 | < .001 | 1980.4±4.8 | 1980.7±6.0 | .068 |
| Age | | 31.9±3.9 | 34.6±9.8 | < .001 | 31.9±3.9 | 31.6±5.2 | .116 |
| Household income The top 50% | | 641 (66.7%) | 14,780 (67.9%) | .461 | 641 (66.7%) | 5,359 (69.8%) | .053 |
| Occupation | White-collar worker | 47 (4.9%) | 5,215 (24.0%) | < .001 | 47 (4.9%) | 1,256 (16.4%) | < .001 |
| | Blue-collar worker | 233 (24.2%) | 6,595 (30.3%) | | 233 (24.2%) | 2,874 (37.4%) | |
| | Unemployed | 681 (70.9%) | 9,960 (45.8%) | | 681 (70.9%) | 3,546 (46.2%) | |
| Weekly working hours | | 17.3±21.1 | 24.1±22.2 | < .001 | 17.3±21.1 | 24.6±22.1 | < .001 |
| Education level Bachelor's degree or more | | 642 (66.8%) | 10,005 (46.0%) | < .001 | 642 (66.8%) | 4,662 (60.7%) | < .001 |
| Residence Metropolitan city | | 444 (46.2%) | 10,268 (47.2%) | .580 | 444 (46.2%) | 4,630 (60.3%) | < .001 |

\* All variables are mean ± SE (standard error) or frequency %; SMD showed the balance of covariate distribution between two matched groups, indicating SMD<0.1 was considered as a sign of good balance.

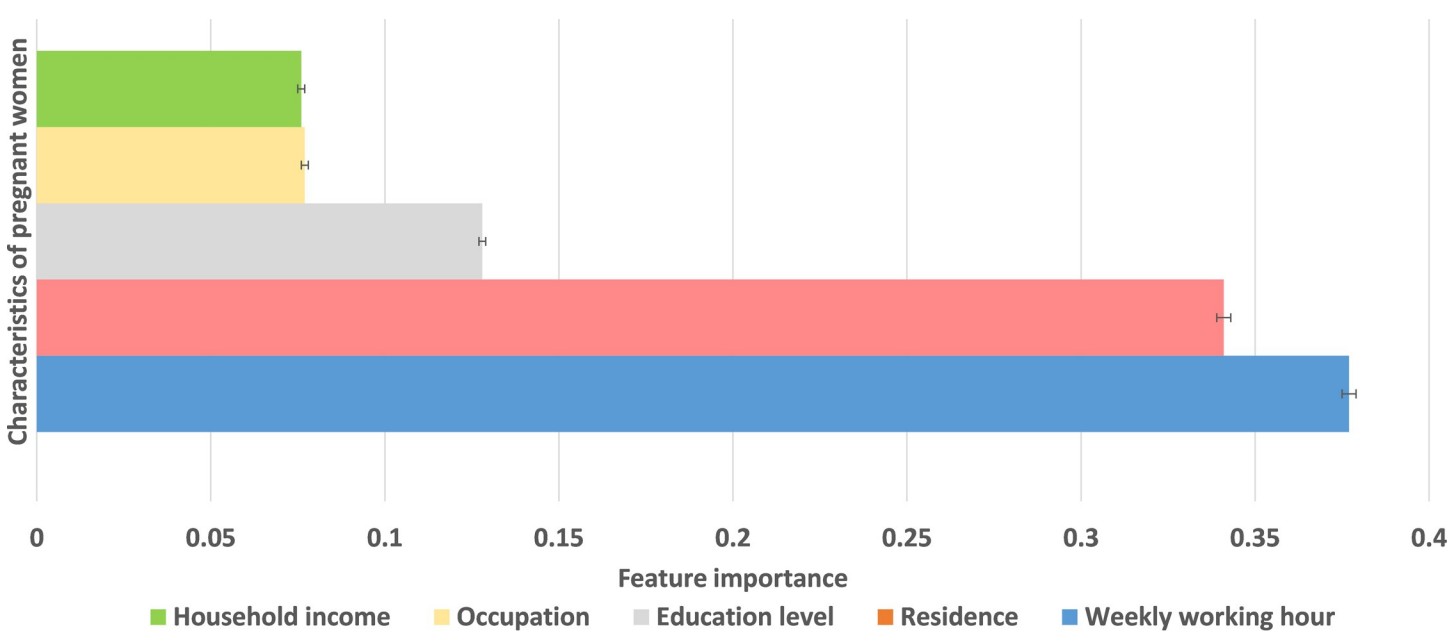

**Fig 3. Feature importance of women's characteristics for pregnancy prediction XGBoost model.**

## Discussion

This study showed that maternal age has increased since 2007 and that a reduction in weekly working hours has been linked to a higher fertility rate. The results of the PS matching analysis indicated that fewer weekly working hours were associated with younger maternal age and a higher pregnancy rate. Furthermore, the findings of the cluster analysis and logistic regression suggest that working between 35 and 45 hours per week increases the likelihood of becoming pregnant compared to those who work less than 35 hours or more than 45 hours per week.

Previous studies worldwide show that long working hours increase the risk of adverse pregnancy outcomes [14]. In the Netherlands, studies have found that women working more than 40 hours per week are at an increased risk of delivering babies with lower birth weights [15].

**Table 4. Current pregnancy and other characteristics of the three clusters for women currently working.**

| Characteristics | | Cluster 1 | Cluster 2 | Cluster 3 | p-value |
|---|---|---|---|---|---|
| | | (N = 10,820) | (N = 1,231) | (N = 69) | |
| **Current pregnancy** | | 274 (2.5%) | 9 (0.7%) | 0 (0.0%) | < .001 |
| **Household income** | | 7,838 (72.4%) | 806 (65.5%) | 45 (65.2%) | < .001 |
| **The top 50%** | | | | | |
| Occupation | **White-collar worker** | 4,309 (39.8%) | 895 (72.7%) | 57 (82.6%) | < .001 |
| | **Blue-collar worker** | 6,485 (59.9%) | 331 (26.9%) | 12 (17.4%) | < .001 |
| | **Unemployed** | 26 (0.2%) | 5 (0.4%) | 0 (0.0%) | < .001 |
| **Weekly working hours** | | 34.4±12.9 | 65.0±7.9 | 96.2±8.8 | < .001 |
| **Education level** | | 5,906 (54.6%) | 386 (31.4%) | 18 (26.1%) | < .001 |
| **Bachelor's degree or more** | | | | | |
| **Residence** | | 5,158 (47.7%) | 503 (40.9%) | 34 (49.3%) | < .001 |
| **Metropolitan city** | | | | | |

* All variables are mean ± SE (standard error) or frequency %

**Table 5. Optimal weekly working hours for pregnancy prediction using logistic model.**

| Weekly working hours | Odd ratio | p-value | AIC |
|---|---|---|---|
| 40–45 | 2.009 (1.581–2.547) | < .001 | 2658.413 |
| 60–65 | 0.197 (0.032–0.616) | 0.022 | 2680.550 |
| 15–20 | 0.307 (0.094–0.723) | 0.019 | 2681.764 |
| 30–35 | 0.516 (0.264–0.901) | 0.032 | 2684.352 |
| 70–75 | 0.188 (0.010–0.838) | 0.095 | 2684.801 |
| 20–25 | 0.532 (0.263–0.951) | 0.051 | 2685.345 |
| 80–85 | 2.653E-6 (0.000–0.021) | 0.961 | 2685.912 |
| 35–40 | 1.450 (1.001–2.040) | 0.040 | 2686.098 |
| 75–80 | 2.655E-4 (0.000–0.041) | 0.962 | 2686.245 |
| 5–10 | 0.441 (0.135–1.044) | 0.106 | 2686.527 |
| 10–15 | 0.549 (0.234–1.083) | 0.120 | 2687.027 |
| 45–50 | 1.250 (0.878–1.734) | 0.196 | 2688.356 |
| 90–95 | 7.247E-6 (0.000–5.975) | 0.965 | 2688.571 |
| 25–30 | 0.722 (0.357–1.293) | 0.317 | 2688.840 |
| 65–70 | 0.529 (0.087–1.668) | 0.373 | 2688.963 |
| 95–100 | 1.972E-5 (0.000–147.5) | 0.963 | 2689.281 |
| 55–60 | 0.737 (0.289–1.523) | 0.463 | 2689.351 |
| 85–90 | 1.973E-5 (0.000–3.932E5) | 0.970 | 2689.517 |
| 0–5 | 1.138 (0.445–2.365) | 0.756 | 2689.850 |
| 50–55 | 0.952 (0.582–1.469) | 0.836 | 2689.900 |

* Minimum AIC was an estimator of prediction error and showed the relative quality of the logistic prediction model.

Additionally, physicians in Japan who work longer hours have experienced frequent pregnancy complications, such as threatened abortions and preterm births [16].

In Korea, the standard working hours per week are typically 40 hours and a 52-hour workweek is implemented in some industries. Moreover, Koreans are known to work longer hours, often unofficially, and by avoiding labor laws. In fact, Korea had the second highest number of working hours among OECD countries in 2022. The effects of these long working hours are considered an important factor in negative health outcomes, such as musculoskeletal symptoms, poor mental health, and decreased kidney function [17–21].

This study suggests that well-educated women in white-collar jobs may face challenges that lead them to leave their careers or be less willing to become pregnant due to pregnancy-related challenges. The study found that pregnant women were more likely to have a bachelor's degree or higher education compared to non-pregnant women (66.8% vs. 60.7%, respectively). However, pregnant women were less likely to work in white-collar positions (4.9% vs. 16.4%, respectively) and even had fewer weekly working hours (average 17.3 hours vs. 24.6 hours). These findings are consistent with previous studies conducted in Korea and Japan, where women of reproductive age often experience intermittent employment due to their responsibilities related to childbearing. Additionally, the inflexible labor market in these countries makes it challenging for women to find similar job opportunities after pregnancy, which may contribute to their hesitation in deciding to become pregnant [22].

In Korea, working women face multiple obstacles when deciding to become pregnant. Women face significant anxiety over the availability and cost of childcare and they desire to advance in their careers [23, 24]. These include concerns about performance evaluation, desire to prioritize childcare, and high levels of education among women in the economy [25, 26].

Although maternity and childcare leave are mandated, many employees are hesitant to use these benefits, as doing so may negatively affect their job performance evaluation [27]. In a study conducted by the UK's Equal Opportunities Commission in 2005, half of pregnant women reported experiencing discrimination or bias due to pregnancy-related issues [28]. Consequently, nearly four out of ten working women in Korea felt compelled to leave their jobs to focus on raising their children [23].

The trend of an increase in the percentage of women with college education or higher, from 13.8% in 1992 to 42.6% in 2016, has resulted in a decreased intention to become pregnant [29]. Highly educated women are more likely to delay marriage and pregnancy not only because of the long time spent in education and training in their field [30, 31], but also due to their desire to continue working. Some choose to work less than they would prefer, whereas others continue working but have fewer children than desired, contributing to Korea's low birth rate [32]. Therefore, it is crucial to address the challenges that women face in balancing work and family responsibilities [33]. In this study, it was observed that pregnant women aged 35 and above showed a higher educational level. However, this difference did not reach statistical significance (66.7% vs. 61.2%, p>0.05). The limited statistical significance could be attributed to the small sample size of pregnant women aged 35 and above, which consisted of only 51 participants in this research. It is anticipated that conducting further analysis with a larger sample size would yield statistically significant results that align with previous studies.

There was no significant difference in household income between pregnant and non-pregnant women. This finding is not aligned with previous studies, which suggest that a higher income is associated with a lower intention of pregnancy [9, 34]. However, Korea has achieved remarkable success in combining rapid economic growth with significant poverty reduction, leading to an improved environment for women. As of 2019, Korea's GDP was 1.8 trillion USD, making it the 10th largest in the world [35], and its' minimum wage was $8.17 as of 2022, making it the 13th highest in the world [36]. According to the results, household income might not be an important factor in planning a pregnancy due to the overall improvement in Korea's economic status.

Our research findings can provide insights and guidance for policymakers aiming to recover the total fertility rate in Korea. The results of this study showed that advanced maternal age has become increasingly prevalent. As maternal age continues to increase, policymakers consider implementing supportive policies to increase access to and affordability of infertility treatment. Multiple prior studies showed that advanced maternal age, defined as being over 35 years of age, is associated with a decline in both ovarian reserve and oocyte competence [37]. Infertility treatments such as in vitro fertilization (IVF) have been proven to increase the possibility of pregnancy [38]. However, the high cost of infertility treatments, owing to the need for highly trained personnel and expensive equipment, often makes it unaffordable for infertile couples [39]. Providing economic support for infertility treatments can be considered as one of the potential solutions to address the issue of low birth rates.

Furthermore, this study has indicated that weekly work hours play a significant role in determining pregnancy, which is associated with work-related burden. Telecommuting could be a suitable solution to this issue as it can alleviate the burden of labor. Previous studies have assessed the association between telecommuting and quality of life, which enables employees to take time off to have a child and spend more time with their families [40, 41]. Since the COVID-19 outbreak, approximately 35% of jobs in Korea have adopted a telecommuting infrastructure [42]. Previous research has suggested that telecommuting leads to increased productivity, reduced work-family conflict [43, 44], and decreased employee turnover rates [45]. However, owing to the recent widespread adoption of telecommuting, only a limited number of studies have examined its potential effects on pregnancy. Further research is needed

to better understand the impact of telecommuting on fertility rates, with the goal of promoting higher fertility rates.

This study has certain limitations. First, the analysis was based on self-reported questionnaires, which may have been subject to recall bias and measurement errors. To improve the accuracy and validity of our findings, future studies should consider using official statistical data from government sources, such as pregnancy rates from the Ministry of Health and weekly working hours from the Ministry of Labor. Additionally, previous research has suggested that the father's characteristics, such as socioeconomic status (e.g. unemployment, level of education, income) and health problems (e.g. previous depressive symptoms, smoking and alcohol intake), may also be associated with pregnancy outcomes [46, 47]. Although the current study focused solely on women's characteristics owing to the absence of paternal information in the KNHANES questionnaire, we hope that future research will include relevant paternal factors in the analysis to provide a more comprehensive understanding of the factors influencing pregnancy outcomes.

From the perspective of the South Korean government, the fall in the birth rate is acknowledged as a serious challenge. Accordingly, a new law was enacted in 2005 to support fertility and mitigate the negative effects of population aging. From 2006 to 2020, a five-year basic plan for increasing fertility and slowing the aging of society was established every five years. Despite the government spending approximately 80 trillion won ($71 billion) over the past decade on these policies, the total fertility rate has not recovered, and the efficacy of this policy has been criticized [48].

Multiple studies have focused on the recovery of the fertility rate, such as changing policy paradigms [49], guaranteeing gender-equal labor policies [50], supporting childcare to relieve the burden on parents [51], decreasing income inequality [52], and assessing attitudes toward welfare [53]. However, this study represents the first statistical analysis investigating the correlation between working hours and fertility in women of childbearing age, suggesting that excessive weekly working hours have a negative effect on the likelihood of pregnancy among women of childbearing-age in Korea.

During a meeting of the Presidential Committee on Aging Society and Population Policy in 2023, President Yoon requested a reevaluation of the low birth rate policy based on scientific evidence. In response to the complex low fertility situation, the government has established and continuously improved the 4th Basic Plan for Low Fertility and Aging Society (2021–2025). We hope that the 4th Basic Plan will address the labor burden of childbearing women and help recover the fertility rate.

## Conclusion

This study examined the association between the pregnancy intention of childbearing-age women and their characteristics, including age, household income, employment rate, occupation, weekly working hours, education level, and residence. The characteristics of childbearing-age women were significantly different among age groups. After adjusting for age and birth year, pregnancy intention was associated with a higher level of education but with a higher proportion of unemployment, a lower proportion of white-collar workers, and fewer weekly working hours. Furthermore, multiple analyses were performed to determine the optimal weekly working hours for childbearing women who intend to get pregnant. The XGBoost model identified weekly working hours as the most important factor influencing pregnancy intention. Cluster analysis on childbearing-age women identified three distinct phenotypes based on their characteristics and found that working 34.4±12.9 hours per week was associated with the highest probability of pregnancy. Furthermore, logistic regression analysis

demonstrated that the odds of pregnancy increased significantly with weekly working hours in the range of 40±5. Overall, these findings suggest that weekly working hours have a significant impact on the intention of childbearing-age women to become pregnant, and that working 34.4±12.9 hours or 40±5 hours per week may be optimal for increasing the probability of pregnancy. These findings provide valuable insights for policymakers, employers, and individuals seeking to support and promote family planning among childbearing-age women in Korea, the country with the world's lowest fertility rate.

## Supporting information

**S1 Checklist. STROBE checklist.**
(DOCX)

## Acknowledgments

The author would like to thank all members and officers of the Presidential Committee on Aging Society and Population Study. The author clarifies that the opinions and content presented in this paper are the sole responsibility of the author and do not reflect the views of the members of the Presidential Committee on Aging Society and Population Study.

## Author Contributions

**Conceptualization:** Taewook Kim.

**Data curation:** Taewook Kim.

**Formal analysis:** Taewook Kim.

**Investigation:** Taewook Kim.

**Methodology:** Taewook Kim.

**Project administration:** Taewook Kim.

**Resources:** Taewook Kim.

**Software:** Taewook Kim.

**Supervision:** Taewook Kim.

**Validation:** Taewook Kim.

**Visualization:** Taewook Kim.

**Writing – original draft:** Taewook Kim.

**Writing – review & editing:** Taewook Kim.

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
