## [Decision Letter · Decision Letter 0]

20 Jun 2023

PONE-D-23-12865The impact of working hours on pregnancy intention in childbearing-age women in Korea, the country with the world’s lowest fertility ratePLOS ONE

Dear Dr. Kim,

Thank you for submitting your manuscript to PLOS ONE. After careful consideration, we feel that it has merit but does not fully meet PLOS ONE’s publication criteria as it currently stands. Therefore, we invite you to submit a revised version of the manuscript that addresses the points raised during the review process.

We look forward to receiving your revised manuscript.

Kind regards,

Ebiere Clara Herbertson, M.Pharm

Academic Editor

PLOS ONE

Reviewers' comments:

Reviewer's Responses to Questions

**Comments to the Author**

1. Is the manuscript technically sound, and do the data support the conclusions?

Reviewer #1: Yes

Reviewer #2: Yes

2. Has the statistical analysis been performed appropriately and rigorously? 

Reviewer #1: Yes

Reviewer #2: Yes

3. Have the authors made all data underlying the findings in their manuscript fully available?

Reviewer #1: Yes

Reviewer #2: Yes

4. Is the manuscript presented in an intelligible fashion and written in standard English?

Reviewer #1: Yes

Reviewer #2: Yes

5. Review Comments to the Author

Reviewer #1: General Overview:

This study delves into the factors influencing pregnancy intentions among women of reproductive age in Korea, utilizing data from the Korean National Health and Nutrition Examination Survey (KNHANES). To analyze the characteristics of pregnant women and identify critical factors related to pregnancy intentions, the authors employed propensity score matching and the XGBoost machine learning model. The study's significant finding indicates that the number of weekly working hours substantially impacts pregnancy intentions. Specifically, women working between 35-45 hours per week exhibit higher odds of desiring pregnancy. Given the Korean context, where individuals tend to work excessively long hours, often surpassing the standard 40-hour workweek, these work patterns may deter or impede pregnancy intentions, affecting the nation's fertility rate.

Consequently, the research urges stricter monitoring of working hours and proposes promoting telecommuting for women in their childbearing years to bolster Korea's fertility rate. Beyond Korea, this study's findings possess broader implications for understanding the relationship between women's work hours and their intentions to conceive worldwide. It can influence workplace policies, particularly in countries with similar work cultures to Korea, by advocating for more flexible work hours or telecommuting to address demographic challenges associated with low birth rates.

Minor Comments:

Introduction:

Line 44: The acronym "OECD" should be defined before its usage.

Method:

Line 85: It is advisable to define "those not of childbearing age – 35,583."

Line 91-94: This section should be under "Ethical Considerations."

Line 96: The current title for this section is not appropriate.

Lines 97-103: This section should be titled "Statistical Analysis."

Lines 106-107: I recommend rewriting this section, as it appears to be mixed with the results. For example, lines 106-107 state, "The descriptive analysis of baseline characteristics and linear regression revealed a significant association among birth year, age, and participant characteristics."

Results:

Line 190-191: Please clarify this sentence: "Household income tended to increase with age, with the highest level observed among participants in the 40–49 age group."

Discussion:

Line 303-304: Reference 22 refers to another study that discusses similar findings. I suggest rephrasing the sentence to convey the message accurately.

Conclusion:

Lines 397-406: The conclusion should shed light on the significant impact of the study findings. Showing the results in this section is not appropriate. Please rephrase/rewrite the conclusion.

Reviewer #2: Manuscript title: Impact of working hours on pregnancy intention in childbearing-age women in Korea, the country with the world’s lowest fertility rate

Comments: the aim of the manuscript was well spelt out, and methods will outlined. The progression in statistical such as: propensity score matching, XGBoost machine learning model,cluster analysis and regression models (pages 12-15) were used with ‘R’. In conclusion, the study was clear of the factors (working hours and expansion of telecommuting) for increasing fertility rate among childbearing-age 42 women in Korea.

There was no major issues, however, minor issues in discussion: Line 192-196 should be well linked to the findings as reported in line 297-302. Previous studies should be inferred as to whether it support your findings or not.

In summary, there is the need to relate your findings to previous research in specific terms.

The use of English is critical . for example you cannot be suggesting policy implementation (as in line 334), and you use MUST (line 335)

Line 334-339 can be rephrased, no need to mention linear regression since you are proposing policy implementation and findings

Line 361-362 not clear (what is father’s situation)

Line 377-378 can be rephrased (However, this is the first statistical analysis study to examine the association between working hours and fertility among women of childbearing-age),

To the best of our understanding …………………………………………in Korea perhaps

Conclusion: Too long, don’t discuss, main finding and suggestion

Recommendation

The study is sound, the context is good, however language can be improved upon. I suggest that authors should revise the language to improve the text.

Strongly recommended with minor corrections

6. PLOS authors have the option to publish the peer review history of their article (what does this mean?). If published, this will include your full peer review and any attached files.

Reviewer #1: **Yes: **Dr. Folahanmi Akinsolu

Reviewer #2: **Yes: **Joseph Anejo-okopi

---

## [Author Response · Author response to Decision Letter 0]

1 Jul 2023

Dear Reviewers

Thank you wholeheartedly for your comments on our manuscript entitled, “The impact of working hours on pregnancy intention in childbearing-age women in Korea, the country with the world’s lowest fertility rate" (PONE-D-23-12865).

We are pleased to submit the revision of our work to the PLOS ONE. We have provided an itemized list of our responses to the reviewer’s comments below. We sincerely appreciate the reviewers' insightful remarks, as their questions and comments have immensely contributed to the improvement of our manuscript.

We are looking forward to hearing from you at your earliest convenience and thank you for considering our manuscript for publication. We believe that this manuscript will be of significant interest to the wide readership of PLOS ONE, and kindly request that you consider it for publication. Please do not hesitate to contact me if you require any further information.

Sincerely,

Taewook Kim, MD

Presidential Committee on Aging Society and Population Study, Seoul Government Complex, Seoul, Republic of Korea

E-mail: ray0601@snu.ac.kr

 

Reviewer #1

Comment 1-1

Line 44: The acronym "OECD" should be defined before its usage.

Answer

We wholeheartedly agree with your comment. We have included the definition of OECD (Organization for Economic Co-operation and Development) the first time it is mentioned in the manuscript.

Changes made in the manuscript:

Introduction (line 44-46)

The fertility rate in OECD (The Organization for Economic Cooperation and Development) countries has decreased from 2.84 in 1970 to 1.59 children per woman in 2020, with some countries experiencing rates below the required level of 2.1 for maintaining a stable population.

Comment 1-2

Line 85: It is advisable to define "those not of childbearing age – 35,583."

Answer

We sincerely appreciate your comment. Given that our study's main objective was to investigate and enhance the total fertility rate in Korea, we specifically targeted women aged 15-49, which corresponds to the definition of the total fertility rate. The total fertility rate is calculated as the sum of fertility rates of women aged 15 to 49 years.

Furthermore, we used the terms "childbearing age" and "reproductive age" interchangeably, with "reproductive age" defined by the World Health Organization (WHO) as "Women of reproductive age (15−49 years) who are married or in-union and have their need for family planning satisfied with modern methods." 

We appreciate your valuable suggestion and have included an additional comment in the article to provide further clarification regarding the definition of childbearing age.

Changes made in the manuscript:

Study population (line 86-88)

We excluded participants with missing or inaccurate data (N = 3,226) and those who were not within the childbearing age range (N = 35,583). Since the definition of the total fertility rate is based on women of reproductive age (15 to 49 years) [3, 7], our study specifically focused on women within this age range.

Comment 1-3

Line 91-94: This section should be under "Ethical Considerations."

Answer

We greatly appreciate your valuable comment. In response, we have included the title "Ethical Considerations" to address the ethical aspects of our study.

Changes made in the manuscript:

Ethical considerations (line 94-96)

Ethical considerations

This study was based on the KNHANES database, which was approved by the Institutional Review Board (IRB) of the KDCA.

Comment 1-4

Line 96: The current title for this section is not appropriate. Lines 97-103: This section should be titled "Statistical Analysis."

Answer

We agree with your comment. To better align with the content of the section, we have revised the title from "Descriptive Analysis" to "Statistical Analysis." Thank you for your valuable suggestion.

Changes made in the manuscript:

Statistical analysis (line 100)

Statistical analysis 

The participants’ baseline characteristics were categorized into the following age groups: 15–19 years, 20–29 years, 30–39 years, and 40–49 years.

Comment 1-5

Lines 106-107: I recommend rewriting this section, as it appears to be mixed with the results. For example, lines 106-107 state, "The descriptive analysis of baseline characteristics and linear regression revealed a significant association among birth year, age, and participant characteristics."

Answer

We greatly appreciate your valuable comments. To enhance clarity, we have included detailed explanations based on the relevant results of this research. These explanations highlight the potential bias introduced by women's age and birth year in the analysis of pregnancy tendencies. Additionally, we emphasize the importance of using the propensity score method to reduce confounding bias from women's age and birth year in this article.

Changes made in the manuscript:

Propensity score matching for characteristics of pregnant women (line 110-115)

The descriptive analysis of baseline characteristics and the linear regression between the age at first childbirth and birth year of women indicated the need to reduce bias for pregnancy tendencies, revealing a significant association among birth year, age, and participant characteristics. For instance, women in the 40-49 age group exhibited higher household income, and women from earlier birth years, indicating the older generation, displayed a tendency to have pregnancies at an earlier age. In addition, previous research has shown that the social environment in which women live can influence

Comment 1-6

Line 190-191: Please clarify this sentence: "Household income tended to increase with age, with the highest level observed among participants in the 40–49 age group."

Answer

We appreciate your comments. I added more descriptions to clear the tendency between household income and age. Our results of descriptive analysis offer a comprehensive overview of various factors concerning women of childbearing age across different age groups. Notably, the results reveal a clear trend of increasing household income with age with participants in the 40–49 age group showing the highest level of household income. This finding suggests that as women enter their late thirties and forties, they tend to have higher household incomes compared to younger age groups. Based on these results, we clearly rephrase the sentence to show a positive association between age and household income among women of childbearing age.

Changes made in the manuscript:

Descriptive analysis (line 197-200)

We analyzed the characteristics of the participants in our study. Table 1 provides an overview of the age distribution of the participants, with four groups analyzed: ages 15–19, 20–29, 30–39, and 40–49. The data showed significant differences among the groups for several factors, including household income, employment rate, occupation, weekly working hours, educational level, and residence (p<.001). Household income showed a tendency to increase as the age group progressed from 15-19 to 40-49 (p<.001). The participants in the 30–39 age group had the highest education level, however, they also had a higher proportion of unemployment and worked fewer weekly hours compared to other age groups (p<.001).

Comment 1-7

Line 303-304: Reference 22 refers to another study that discusses similar findings. I suggest rephrasing the sentence to convey the message accurately.

Answer

Thank you for your valuable comments regarding the rephrasing of the sentence. We have thoroughly reviewed reference 22 and revised the paragraph accordingly to ensure accurate conveyance of the message. We believe that these revisions clearly reflect the findings of our study and the results of reference 22. 

Once again, we appreciate your comments, which has improved the clarity and accuracy of our manuscript.

Changes made in the manuscript:

Discussion (line 310 -320)

This study suggests that well-educated women in white-collar jobs may face challenges that lead them to leave their careers or be less willing to become pregnant due to pregnancy-related challenges. The study found that pregnant women were more likely to have a bachelor's degree or higher education compared to non-pregnant women (66.8% vs. 60.7%, respectively). However, pregnant women were less likely to work in white-collar positions (4.9% vs. 16.4%, respectively) and even had fewer weekly working hours (average 17.3 hours vs. 24.6 hours). These findings are consistent with previous studies conducted in Korea and Japan, where women of reproductive age often experience intermittent employment due to their responsibilities related to childbearing. Additionally, the inflexible labor market in these countries makes it challenging for women to find similar job opportunities after pregnancy, which may contribute to their hesitation in deciding to become pregnant [22].

Comment 1-8

Lines 397-406: The conclusion should shed light on the significant impact of the study findings. Showing the results in this section is not appropriate. Please rephrase/rewrite the conclusion 

Answer

We wholeheartedly agree with your comment. Considering your valuable feedback, we have minimized the emphasis on numerical results and instead focused on providing a deeper interpretation of the significance of the research findings within a societal context. We sincerely appreciate your insightful comments.

Changes made in the manuscript:

Conclusion (line 421-428)

Cluster analysis on childbearing-age women identified three distinct phenotypes based on their characteristics and found that working 34.4±12.9 hours per week was associated with the highest probability of pregnancy. Furthermore, logistic regression analysis demonstrated that the odds of pregnancy increased significantly with weekly working hours in the range of 35–45. Overall, these findings suggest that weekly working hours have a significant impact on the intention of childbearing-age women to become pregnant, and that working 34.4±12.9 hours or 40±5 hours per week may be optimal for increasing the probability of pregnancy. These findings provide valuable insights for policymakers, employers, and individuals seeking to support and promote family planning among childbearing-age women in Korea, the country with the world’s lowest fertility rate.

Reviewer #2

Comment 2-1

Line 192-196 should be well linked to the findings as reported in line 297-302. Previous studies should be inferred as to whether it support your findings or not.

Answer

We completely agree with your comment. We have incorporated more detailed explanations based on previous studies and have assessed whether they support our findings or not. 

Firstly, we have included a discussion on education level and weekly working hours. The findings of our study align with previous research in this regard. 

Furthermore, we have delved deeper into the relationship between education level and delayed pregnancy. Previous studies have suggested a correlation between higher education levels and delayed pregnancy. However, the findings of our study reveal a similar trend, although lacking statistical significance. We attribute this to the small sample size of women aged 35 and above who became pregnant, which limited our ability to obtain statistical significance. 

Moreover, we have extensively discussed the association between high-income level and the tendency for pregnancy. In contrast to previous studies, our findings did not reveal a significant association between a higher income level and a stronger inclination for pregnancy. To clearly demonstrate this result, we have incorporated additional sentences based on previous studies, considering the enhanced economic status in Korea.

Deeply thanks for pointing out the need for more detailed explanations and discussions. We have considered your feedback and have expanded on these aspects in our study.

Changes made in the manuscript:

Discussion (line 310-320, 338-343, 344-346)

This study suggests that well-educated women in white-collar jobs may feel compelled to leave their careers due to pregnancy-related challenges. While pregnant women were more likely to have attained a bachelor’s degree or completed higher education than non-pregnant women (66.8% vs. 60.7%, respectively), they were less likely to work in white-collar positions (4.9% vs. 16.4%, respectively). The difference between a high education level (66.8% vs 60.7%) and fewer weekly working hours (average 17.3 hours vs 24.6 hours) in pregnant women suggests that pregnancy-related challenges actually lead women to quit their jobs. These findings align with previous studies. In Korea and Japan, due to the inflexible labor market, women are more likely to have intermittent employment due to childbearing responsibilities [22].

The trend of an increase in the percentage of women with college education or higher, from 13.8% in 1992 to 42.6% in 2016, has resulted in a decreased intention to become pregnant [31]. Highly educated women are more likely to delay marriage and pregnancy not only because of the long time spent in education and training in their field [29, 30], but also due to their desire to continue working. Some choose to work less than they would prefer, whereas others continue working but have fewer children than desired, contributing to Korea’s low birth rate [32]. Therefore, it is crucial to address the challenges that women face in balancing work and family responsibilities [33]. In this study, it was observed that pregnant women aged 35 and above showed a higher educational level. However, this difference did not reach statistical significance (66.7% vs. 61.2%, p>0.05). The limited statistical significance could be attributed to the small sample size of pregnant women aged 35 and above, which consisted of only 51 participants in this research. It is anticipated that conducting further analysis with a larger sample size would yield statistically significant results that align with previous studies.

There was no significant difference in household income between pregnant and non-pregnant women. This finding is not aligned with previous studies, which suggest that a higher income is associated with a lower intention of pregnancy [9, 34]. However, Korea has achieved remarkable success in combining rapid economic growth with significant poverty reduction, leading to an improved environment for women. As of 2019, Korea’s GDP was 1.8 trillion USD, making it the 10th largest in the world [35], and its’ minimum wage was $8.17 as of 2022, making it the 13th highest in the world [36]. According to the results, household income might not be an important factor in planning a pregnancy due to the overall improvement in Korea’s economic status.

Reviewer #2

Comment 2-2

The use of English is critical, for example you cannot be suggesting policy implementation (as in line 334), and you use MUST (line 335) Line 334-339 can be rephrased, no need to mention linear regression since you are proposing policy implementation and findings.

Answer

We absolutely agree with your comment. We changed the words you mentioned, and rephrased the paragraph to provide a clear proposal for policy implementation. Rather than proposing a policy, we emphasized the potential of our research to be helpful in informing policy decisions. Thank you for your valuable feedback.

Changes made in the manuscript:

Discussion (line 353-358)

Our research findings can provide insights and guidance for policymakers aiming to recover the total fertility rate in Korea. The results of this study showed that advanced maternal age has become increasingly prevalent. As maternal age continues to increase, policymakers consider implementing supportive policies to increase access to and affordability of infertility treatment. Multiple prior studies showed that advanced maternal age, defined as being over 35 years of age, is associated with a decline in both ovarian reserve and oocyte competence [37]. Infertility treatments such as in vitro fertilization (IVF) have been proven to increase the possibility of pregnancy [38]. However, the high cost of infertility treatments, owing to the need for highly trained personnel and expensive equipment, often makes it unaffordable for infertile couples [39]. Providing economic support for infertility treatments can be considered as one of the potential solutions to address the issue of low birth rates.

Reviewer #2

Comment 2-3

Line 361-362 not clear (what is father’s situation)

Answer

We appreciate your comment. To clarify, we have made the change from using the word "situation" to "characteristics," and we have included additional descriptions and examples of the father's characteristics based on references 46 and 47.

Changes made in the manuscript:

Discussion (line 380-383)

This study has certain limitations. First, the analysis was based on self-reported questionnaires, which may have been subject to recall bias and measurement errors. To improve the accuracy and validity of our findings, future studies should consider using official statistical data from government sources, such as pregnancy rates from the Ministry of Health and weekly working hours from the Ministry of Labor. Additionally, previous research has suggested that the father’s characteristics, such as socioeconomic status (e.g. unemployment, level of education, income) and health problems (e.g. previous depressive symptoms, smoking and alcohol intake), may also be associated with pregnancy outcomes [46, 47]. Although the current study focused solely on women’s characteristics owing to the absence of paternal information in the KNHANES questionnaire, we hope that future research will include relevant paternal factors in the analysis to provide a more comprehensive understanding of the factors influencing pregnancy outcomes.

Reviewer #2

Comment 2-4

Line 377-378 can be rephrased (However, this is the first statistical analysis study to examine the association between working hours and fertility among women of childbearing-age),

Answer

We greatly appreciate your insightful comment. In response to your feedback, we have rephrased the mentioned sentences to highlight the significance of our study as the first statistical analysis conducted on women of childbearing age. Thank you for your valuable input.

Changes made in the manuscript:

Discussion (line 398-401)

Multiple studies have focused on the recovery of the fertility rate, such as changing policy paradigms [49], guaranteeing gender-equal labor policies [50], supporting childcare to relieve the burden on parents [51], decreasing income inequality [52], and assessing attitudes toward welfare [53]. However, this study represents the first statistical analysis investigating the correlation between working hours and fertility in women of childbearing age, suggesting that excessive weekly working hours have a negative effect on the likelihood of pregnancy among women of childbearing-age in Korea.

---

## [Editor Report · Decision Letter 1]

4 Jul 2023

The impact of working hours on pregnancy intention in childbearing-age women in Korea, the country with the world’s lowest fertility rate

PONE-D-23-12865R1

Dear Dr. Taewook Kim

We’re pleased to inform you that your manuscript has been judged scientifically suitable for publication and will be formally accepted for publication once it meets all outstanding technical requirements.

Kind regards,

Ebiere Clara Herbertson, M.Pharm

Academic Editor

PLOS ONE
---

## [Editor Report · Acceptance letter]

10 Jul 2023

PONE-D-23-12865R1 

The impact of working hours on pregnancy intention in childbearing-age women in Korea, the country with the world’s lowest fertility rate 

Dear Dr. Kim:

I'm pleased to inform you that your manuscript has been deemed suitable for publication in PLOS ONE. Congratulations! Your manuscript is now with our production department. 

Kind regards, 

on behalf of

Dr. Ebiere Clara Herbertson 

Academic Editor

PLOS ONE